# Parenteral Nutrition, Inflammatory Bowel Disease, and Gut Barrier: An Intricate Plot

**DOI:** 10.3390/nu16142288

**Published:** 2024-07-17

**Authors:** Carlo Covello, Guia Becherucci, Federica Di Vincenzo, Angelo Del Gaudio, Marco Pizzoferrato, Giovanni Cammarota, Antonio Gasbarrini, Franco Scaldaferri, Maria Chiara Mentella

**Affiliations:** 1Gastroenterology Department, Centro di Malattie dell’Apparato Digerente (CEMAD), Center for Diagnosis and Treatment of Digestive Diseases, Fondazione Policlinico Universitario A. Gemelli IRCCS, 00168 Rome, Italy; covellocarlo@gmail.com (C.C.); federica.divincenzo30@gmail.com (F.D.V.); delgaudioangelo@gmail.com (A.D.G.); antonio.gasbarrini@unicatt.it (A.G.); 2UOS Malattie Infiammatorie Croniche Intestinali, Centro di Malattie dell’Apparato Digerente (CEMAD), Medicina Interna e Gastroenterologia, Fondazione Policlinico Universitario A. Gemelli IRCCS, 00168 Rome, Italy; guia.becherucci01@icatt.it (G.B.); franco.scaldaferri@policlinicogemelli.it (F.S.); 3UOC Gastroenterologia, Dipartimento di Scienze Mediche e Chirurgiche, Fondazione Policlinico Universitario Agostino Gemelli IRCCS, 00168 Rome, Italy; marco.pizzoferrato@policlinicogemelli.it (M.P.); giovanni.cammarota@unicatt.it (G.C.); 4Department of Translational Medicine and Surgery, Università Cattolica del Sacro Cuore, 00168 Rome, Italy; 5UOC di Nutrizione Clinica, Dipartimento Scienze Mediche e Chirurgiche Addominali ed Endocrino-Metaboliche, Fondazione Policlinico Universitario A. Gemelli IRCCS, 00168 Rome, Italy

**Keywords:** parenteral nutrition, gut microbiota, gut barrier, intestinal barrier, intestinal permeability, dysbiosis, malnutrition, inflammatory bowel disease, short bowel syndrome

## Abstract

Malnutrition poses a critical challenge in inflammatory bowel disease, with the potential to detrimentally impact medical treatment, surgical outcomes, and general well-being. Parenteral nutrition is crucial in certain clinical scenarios, such as with patients suffering from short bowel syndrome, intestinal insufficiency, high-yielding gastrointestinal fistula, or complete small bowel obstruction, to effectively manage malnutrition. Nevertheless, research over the years has attempted to define the potential effects of parenteral nutrition on the intestinal barrier and the composition of the gut microbiota. In this narrative review, we have gathered and analyzed findings from both preclinical and clinical studies on this topic. Based on existing evidence, there is a clear correlation between short- and long-term parenteral nutrition and negative effects on the intestinal system. These include mucosal atrophic damage and immunological and neuroendocrine dysregulation, as well as alterations in gut barrier permeability and microbiota composition. However, the mechanistic role of these changes in inflammatory bowel disease remains unclear. Therefore, further research is necessary to effectively address the numerous gaps and unanswered questions pertaining to these issues.

## 1. Introduction

Inflammatory bowel diseases (IBDs) are a group of chronic and recurring inflammatory bowel disorders that have relatively low mortality rates. These disorders, including ulcerative colitis (UC), Crohn’s disease (CD), and unclassified IBD (IBD-U), commonly affect the small intestine and colon. However, they differ in terms of the location and severity of the injury [1]. Both incidence and prevalence of IBDs are on the rise globally and are expected to continue increasing in the coming years, particularly in industrialized countries [2].

The intestinal microbial community appears to be crucially involved in the pathogenesis of the diseases, as IBD predominantly affects the most colonized intestinal segments, the distal ileum and colon [3]. In the context of inflammatory bowel disease (IBD), the function of the epithelial barrier within the intestine is compromised. It is well established that the intestinal barrier is a critical player in regulating inflammatory responses to the microbiota. This barrier is characterized by a finely tuned network of immune mechanisms that recognize and tolerate microbes [4]. Early investigations showed a reduction in tight junction filaments in both ulcerative colitis (UC) and Crohn’s disease (CD) [5,6]. According to recent research, it has been posited that changes to the permeability of the epithelial barrier may compromise the host’s ability to protect against microbial invasion and proliferation. This, in turn, may result in a greater degree of interaction between bacteria and mucosal immune cells, triggering abnormal immune responses that can give rise to chronic inflammation of the gut [4].

Among environmental factors, dietary habits and dietary-nutritional setting appear to play an important role in the pathogenesis and management of IBD through their ability to modulate the composition of the gut microbiota [7]. Certain components of foods induce changes in the composition and function of the gut microbiota, which interacts with the intestinal epithelium and the mucosal immune system and maintains intestinal homeostasis in a healthy state [8]. In this context, parenteral nutrition (PN), defined as the process of intravenous administration of nutrition to patients who are unable to meet their nutritional needs through enteral feeding, appears to play a role in influencing the barrier and the gut microbiota. For IBD patients, parenteral nutrition is an important strategy that should be considered in specific situations, such as cases of short bowel syndrome, high-flow intestinal fistula, or small bowel obstruction. The use of parenteral nutrition is crucial to ensure proper nutritional intake and prevent malnutrition, which may lead to worsened outcomes [9]. Moreover, empirical data demonstrate that approximately 25% of patients admitted with UC and CD receive parenteral nutrition (PN), compared to 6% of patients without UC and CD [9]. This underscores the significant role of PN as a therapeutic intervention for these patients.

Despite previous evidence suggesting the negative impact of parenteral nutrition on gut barrier and microbiota, the actual impact of PN on IBD patients experiencing disrupted gut health remains unclear. The aim of this paper is to provide an overview of the potential impacts of parenteral nutrition on the intestinal system, with a focus on the possible risks associated with chronic inflammatory bowel disease. Specifically, this narrative review highlights the potential of parenteral nutrition for harm to the intestinal barrier and the disruption of the microbiota. In this paper, we will delve into the clinical and nutritional applications of parenteral nutrition in IBD. We will explore the potential impact of this treatment approach on the gut barrier and microbiota and identify possible challenges that may arise in future research.

## 2. Parenteral Nutrition in Inflammatory Bowel Disease

The selection of nutritional therapy for IBD patients is a multifactorial process that requires careful consideration of various aspects. These include the patient’s feeding capacity, the absorption ability of the gastrointestinal tract, the patient’s nutritional and inflammatory status, and the therapeutic goals [8]. Generally, the preferred means of nutritional support for most disease states is enteral nutrition (EN) due to lower rates of related complications and the preservation of gastrointestinal function. EN is considered a more natural way to promote organ function, and it helps with mucosal healing, maintaining remission, and supporting surgical treatment [10]. However, several contraindications may preclude the use of this approach. Intestinal obstruction, severe shock, intestinal ischemia, high-output fistula, and severe intestinal hemorrhage are all conditions that make it inadvisable to use enteral nutrition. In these cases, parenteral nutrition (PN) may be required for days or weeks until the gastrointestinal tract function is restored [11].

The notion of bowel rest, which is commonly observed in clinical practice, has been found to lack scientific evidence and may pose potential risks when associated with PN infusion. In a systematic review of randomized trials assessing the efficacy of PN support in patients with various disease states, including Crohn’s disease, insufficient data were identified to support the premise that bowel rest is necessary to induce remission of Crohn’s disease [12]. Therefore, it is important to carefully evaluate each patient’s condition before deciding whether to use EN or PN [8,13]. The illustration below depicts a practical algorithm for nutritional support management for inflammatory bowel disease (Figure 1).

Parenteral nutrition involves delivering a precisely balanced combination of macronutrients directly into the bloodstream through a central or peripheral vein [8]. According to the European Society of Clinical Nutrition and Metabolism (ESPEN) guidelines, it is advised for IBD patients when they are unable to consume oral or enteral nutrition for a period of 7–10 days. Specifically, it is recommended when patients cannot meet more than 60% of their nutritional needs through oral or enteral nutrition [14]. This is common in cases of chronic intestinal obstruction, gastrointestinal tract dysfunction, or CD patients with short bowel syndrome (SBS) in case of intractable vomiting and/or diarrhea [11]. While there is no single recommended PN composition for all IBD patients, it is important to ensure that PN support provides sufficient energy intake. This can be estimated using indirect calorimetry, which is a complex and costly method, or using predictive formulas [15].

This nutritional approach is often used and is also recommended when it is necessary to supplement oral or enteral nutrition with intravenous support [14] as a therapeutic weapon against malnutrition, a condition that, in IBD, has a worldwide prevalence ranging from 5 to 92% [16,17]. IBD patients are more prone to malnutrition due to various factors, including reduced food intake, intestinal malabsorption, increased energy requirements due to hypercatabolism, and chronic loss of protein in the stool [17]. It is essential to consistently identify malnourished patients or those at ‘severe metabolic nutritional risk’ based on the criteria outlined by the ESPEN working group. This includes weight loss of >10–15% within six months, a body mass index (BMI) < 18.5 kg/m^2^, a Nutritional Risk Screening 2002 score of >5, or a serum albumin level < 30 g/L (without signs of liver or kidney dysfunction) [8]. In this scenario, parenteral nutrition, whether combined or total (TPN), is essential in improving the metabolic conditions of patients [18] through, for example, the positive impact it has on body weight [19] and albumin levels [20].

IBD may give rise to a debilitating condition called intestinal failure (IF), which is defined as “the reduction of intestinal function below the minimum necessary for absorption of macronutrients and/or water and electrolytes, such that intravenous supplementation is required to maintain health and/or growth” [21]. Chronic intestinal failure (CIF) is a long-term condition necessitating extended intravenous support, often leading to the use of home parenteral nutrition (HPN) for months or years [22].

Thus, intestinal failure is a potential complication for IBD patients resulting from a variety of pathological factors. In this context, large bowel resections can lead to the development of a condition known as short bowel syndrome (SBS). SBS is the most common cause of intestinal failure and is characterized by malabsorption in the intestine and a functional length of the small intestine that is less than 200 cm [22]. Notably, the volume and type of intravenous fluid administration are predictors of the severity of chronic SBS-IF, as they are independently associated with patient outcomes and major complications of IF. Patients with chronic SBS-IF requiring PN support have more severe SBS-IF than those receiving only intravenous fluids and electrolytes. In addition, patients requiring low-volume PN (<1 L/day) are more likely to be weaned off supplementation and also have a lower likelihood of death and major complications than patients requiring higher support [23]. In cases of CIF, it is important to assess the presence of electrolyte and/or metabolic nutritional imbalances in patients. If the patient presents a normal balance and if the clinical–nutritional assessment allows, a gradual transition to enteral or oral nutrition can be considered. Conversely, if electrolyte and/or metabolic nutritional imbalances are evident at the weaning attempt, parenteral nutrition should be continued. At this point, the introduction of an intestinal growth factor that is a recombinant GLP-2 analog, teduglutide, may be considered. Teduglutide serves to improve nutritional and hydration status by restoring small intestinal epithelial function, thereby facilitating sodium-coupled glucose transport and water absorption [24]. Clinical studies show that treatment with teduglutide should not be started until the patient can be reasonably assumed to be stable after a period of postoperative bowel adaptation, which usually occurs 12 months after the last bowel resection but can range from 6 months to 36 months [25].

PN has also proven its effectiveness in the preoperative field by restoring energy and protein stores, reducing micronutrient deficiencies, and preventing post-surgical malnutrition. Furthermore, the use of preoperative PN decreases complications in the postoperative period in patients with IBD [26]. In the case of elective surgery in patients with severe malnutrition, guidelines recommend delaying surgery by 7–14 days [27]. PN, even when combined with enteral support if feasible, can facilitate the swift enhancement of nutritional status during this preoperative period (Figure 1). During the postoperative period, it is recommended to start EN within 24 h after surgery [28]. If contraindications for enteral nutrition are present and nutrition therapy is warranted, prompt administration of parenteral nutrition is advised [28]. The duration of this period may vary depending on the type of surgery and the patient’s condition [15,27,29].

Parenteral nutrition entails potential complications that are broadly divided into catheter-related and NP-related complications. Catheter-related complications primarily pertain to infections that may arise in the context of central venous access and may also include pneumothorax, hemorrhage, gas embolism, phlebitis, venous thrombosis, and sepsis [30,31]. In contrast, NP-related complications include metabolic disorders, liver failure, hypoglycemia, thrombosis, hypertriglyceridemia, hypokalemia, hepatic damage, and reduced quality of life [32,33].

## 3. Human Gut Barrier and Microbiota

### 3.1. Composition and Function of a Healthy Gut Barrier

The human gastrointestinal tract is a complex and sophisticated system that comprises a diverse community of microbes that communicate with each other and the host. The gut barrier is crucial to maintaining human health, serving as a critical interface between the external and internal environments of the body [34]. This barrier, together with the intricate epithelial and immunological systems, is responsible for ensuring intestinal homeostasis. The gut barrier serves a dual role, namely a “functional” role, allowing the absorption of nutrients and fluids, and an “anatomical” role, preventing the passage of harmful substances, such as toxins and pathogenic bacteria, from crossing the intestinal epithelium and infiltrating the human organism [35]. The presence of a dysfunctional barrier can lead to a decline in an individual’s health status. This condition has been observed in patients with inflammatory bowel disease, as well as those with extraintestinal conditions such as obesity, type II diabetes, and Parkinson’s disease [34].

The use of parenteral nutrition has been correlated with disturbances in the integrity of the intestinal barrier. These disturbances encompass the epithelial, immunological, and microbiological components of the barrier.

The epithelial barrier encompasses enterocytes and intercellular junctions, commonly known as tight junctions (TJs). TJs establish the physical intercellular barrier, delineating tissue spaces and governing the selective movement of solutes through the epithelium [36]. TJ transmembrane proteins, such as claudins, junctional adhesion molecules (JAMs), and MARVEL tight junction-associated proteins, also control the transcellular transport of ions and solutes [37]. In the mucosa, enterocytes, enteroendocrine cells, mucus-secreting goblet cells, and microfold cells (M cells) collaborate to maintain a barrier function [38]. The intestinal epithelium undergoes renewal every 3–5 days through the activity of stem cells located within the intestinal crypts [39], ensuring the continuity and integrity of the epithelial barrier [40].

The intestinal epithelium is shielded by a protective mucous layer containing mucins, which are highly glycosylated glycoproteins originating from specialized epithelial cells known as goblet cells [41]. This mucus layer serves to safeguard and lubricate the intestine while engaging with the microbiota [42] and ensnaring immunoglobulins and antimicrobial peptides within its viscous composition [43]. Antimicrobial peptides (e.g., lysozyme, α-defensins and β-defensins, lectins) secreted by Paneth cells [44] bind to the mucus surface and interact with the microbiota, inhibiting colonization by pathogenic microbes in favor of commensal ones [45]. Paneth cell function is partly regulated by Th2 cytokines (IL-4, IL-9, and IL-13), as well as by the hormone GLP-2 and other intestinotrophic agents [46,47,48]. “Mucosal” immunoglobulins (IgAs), which are manufactured by plasma cells in the lamina propria, can neutralize bacterial toxins and simultaneously ensnare bacteria through opsonization mechanisms [37,49] while also facilitating immune tolerance processes [50]. The health of the mucosal barrier is intricately connected to the gut-associated lymphoid tissue (GALT) and its diverse cellular components, including mucosa-associated T-invariant cells, γδT cells, innate lymphoid cells, and plasmacytoid dendritic cells [51]. Within multifollicular lymphoid tissues like Peyer’s patches (PPs) in the intestine and isolated lymphoid follicles (ILFs), GALT immune cells play a crucial role in maintaining a delicate balance between tolerance to beneficial microbes and nutrients and protection against harmful pathogens; this promotes barrier integrity and ensures protective immunity [52].

The epithelial barrier and the immunological barrier are, in turn, connected to the dynamic microorganism communities of the gut microbiota through a wide range of intricate mechanisms that are currently being studied in the fields of metaproteomics, metatracriptomics, and metabolomics. Indeed, the gut barrier can also be considered ‘‘a microbiological barrier” colonized by almost 10^13^ bacterial cells, together with more than 250 species of viruses, fungi, and archaea, constituting the so-called ‘‘gut microbiota”. These interacting microorganisms establish a mutualistic relationship with the host and significantly influence numerous physiological processes by maintaining a state of dynamic equilibrium [53]. Predominantly composed of five bacterial phyla, namely Firmicutes (60% to 80%), Bacteroidetes (20% to 40%), Verrucomicrobia, Actinobacteria, and, to a lesser extent, Proteobacteria, along with the archaic phylum Euryarchaeota [54], the gut microbiota plays a key role in energy absorption, polysaccharide digestion, short-chain fatty acid (SCFA) production, bile acid (BA) metabolism, and amino acid and vitamin synthesis (e.g., folate, vitamin K, thiamine, biotin, riboflavin, and pantothenic acid), as well as the maintenance of intestinal barrier integrity, protection from exogenous pathogens, maturation of the host intestinal immune system, and the metabolism of xenobiotics [55,56,57]. The term “dysbiosis” is used to describe a state characterized by decreased microbial variety and diversity, alterations in the Firmicutes/Bacteroidetes ratio, proliferation of lipopolysaccharide-producing Gram-negative Proteobacteria (LPS), and heightened intestinal permeability [58,59,60,61]. In dysbiotic conditions, there is an excessive presence of bacteria and their byproducts crossing the intestinal barrier, causing an overproduction of proinflammatory cytokines. This results in epithelial damage and chronic inflammation, which is commonly seen in chronic inflammatory diseases and short bowel syndrome [62].

### 3.2. Specific Barrier Changes in Inflammatory Bowel Disease and Short Bowel Syndrome

Inflammatory bowel diseases constitute a group of disorders resulting from immune system dysregulation and leading to intestinal inflammation and microbial dysbiosis [63]. Multiple lines of evidence, such as the clinical observation that IBD patients can respond to antibiotic therapy, the susceptibility to inflammation of anatomical regions with relative fecal stasis, like the terminal ileum and the rectum, and the effectiveness of fecal diversion as a treatment for CD, suggest that the gut microbiota plays a significant role in driving and promoting intestinal inflammation [64,65,66,67,68]. Notably, a loss of gut microbial diversity, indicative of dysbiosis, is commonly observed in IBD patients, with increased intestinal permeability also noted in asymptomatic individuals years before IBD onset [69]. A large cohort study involving 1420 first-degree healthy relatives of CD patients revealed that elevated intestinal permeability, assessed via the urinary fractional excretion of the lactulose/mannitol ratio (LMR), serves as a predictive marker for CD onset several years in advance [69].

This heightened intestinal permeability often correlates with disease severity in IBD patients but also persists during remission periods. Notably, it is primarily attributed to abnormalities in tight junction proteins, while severe mucosal damage during disease activity further compromises barrier integrity, leading to uncontrolled luminal content leakage [69]. For instance, TNF-α, in conjunction with IL-13, has been implicated in tight junction disruption and in the induction of epithelial-cell apoptosis (Figure 2). Despite the pivotal role of gut barrier function in IBD, our understanding of the underlying regulatory mechanisms remains limited, largely due to the paucity of human data, with most insights derived from animal studies demonstrating how microbiota alterations can disrupt intestinal barrier homeostasis [70].

The healthy intestinal barrier is composed of multiple layers. Polarized, columnar epithelial cells form the mucosal layer, which includes enterocytes and other specialized cell types. A protective layer of mucus containing antimicrobial peptides and immunoglobulin A (IgA) shields the epithelial barrier from microbiota and pathogens. A balanced, diverse eubiotic microbiota and its metabolites, including short-chain fatty acids (SCFAs), play a crucial role in maintaining mucosal integrity. Disruptions in the mucosal membrane, altered expression of tight junction (TJ) proteins, and immune cell infiltration are prominent features in inflammatory bowel disease (IBD), leading to intestinal “leakiness” and increased intestinal permeability. This heightened intestinal permeability allows noxious stimuli/pathogens to enter the lamina propria, triggering a proinflammatory immune response via the activation of T-helper (Th) cells-17, Th-1, and Th-2, and the inhibition of T-regulatory (Treg) lymphocytes. This proinflammatory cascade further recruits other immune cells through the production of proinflammatory cytokines and chemokines, thereby perpetuating the inflammatory cycle and leading to systemic inflammation via the bloodstream. In patients with short bowel syndrome (SBS), an abundance of unabsorbed nutrients in the intestinal lumen leads to bacterial overgrowth, with the composition varying depending on the involved intestinal tract. This results in both the development of intestinal dysbiosis and secondarily altered intestinal permeability, as well as the overproduction of bacterial metabolites, such as lactic acid, with associated systemic consequences.

Compared to healthy individuals, the gut microbiota of IBD patients exhibits marked reductions in microbial diversity, characterized by significant decreases in Firmicutes and *Faecalibacterium prausnitzii*, alongside notable increases in members of Proteobacteria phylum, such as Enterobacteriaceae, including adherent-invasive *Escherichia coli*, and mucolytic bacteria such as *Ruminococcus gnavus* and *Ruminococcus torques* [63]. Some bacteria, such as *Mycobacterium avium* subsp. *Paratuberculosis*, have been investigated as a potential cause of CD due to their ability to cause chronic granulomatous enteritis in animal models [71,72,73,74], while *Fusobacterium nucleatum* has been associated with the development of cancer in ulcerative colitis patients, although a clear cause–effect relationship cannot be proven [75,76].

Experimental animal studies further support the involvement of gut microbiota in IBD pathophysiology. The transplantation of disease-associated microbiota transmitted the clinical symptoms of a CD-like ileitis to germ-free TNF (deltaARE) mice [77]. Similarly, studies by C. Eftychi et al. demonstrated that specific symbiotic flora combinations could induce colitis in mice with impaired intestinal barrier function [78]. Additionally, the intestinal microbiota’s role is under investigation, with the fungal fraction of the gut possibly implicated in CD, as suggested by the anti-Saccharomyces cerevisiae antibodies (ASCA) biomarker of CD [79]. Furthermore, gut microbial metabolites play critical roles in IBD pathogenesis. Short-chain fatty acids (SCFAs), typically depleted in IBD patients, exert diverse effects on mucosal immunity by promoting B-cell development and maintaining mucosal integrity via inflammasome activation and IL-18 production [80]. Bile acids exhibit immunomodulatory properties by directly stimulating FXR, which mediates anti-inflammatory effects and protects against chemically induced colitis [81]. These findings underscore the significant contribution of gut microbiota to IBD onset and progression, although the specific microbial populations and mechanisms involved warrant further investigation.

In IBD, SBS has been notoriously associated with the alteration of the gut microbiota in the context of metabolic acidosis. Due to the increased delivery of carbohydrates to the colon, SBS is characterized by a considerably increased D-lactate production by the intestinal microbiota in the colon [82]. Indeed, previous studies revealed that total fecal Bacteroidetes were significantly decreased, while the ratio of lactobacilli, particularly of those that were D-lactate producers, to other bacteria was elevated [82,83,84,85]. In this scenario, the alterations of the intestinal microbiota appear to be adaptive and secondary to the SBS state, aiming to increase energy absorption in the colon, albeit with the negative effect of acidosis [86].

When dysbiosis occurs in SBS, it often leads to small intestinal bacterial overgrowth (SIBO) [87,88]. SIBO is characterized by having >10^5^ CFU/mL of bacteria or >10^3^ CFU/mL of bacterial species typically found in the colon [88]. SIBO can manifest with various symptoms, including feeding intolerance, bloating, and abdominal pain, making the development of effective strategies crucial for managing patients with SIBO [89,90]. This condition involves changes in intestinal permeability that occur after the initial damage. This can lead to the overproduction of bacterial metabolites, such as lactic acid, which can have systemic consequences.

It is interesting to note that the gastrointestinal tract is split into different sections, each with its own unique composition of gut bacteria, depending on its location. For instance, Helicobacter, Lactobacillus, and Veillonela are the most abundant in the proximal intestine, whereas Streptococcaceae, Actinobacteria, and Bacilli are increased in the duodenum, jejunum, and ileum, respectively. On the other hand, a high population of Lachnospiraceae and Bacteriodetes is found in the colon [91,92]. As a result, depending on the site and extent of short bowel syndrome (SBS), a specific pattern of dysbiosis of the gut microbiome can be identified. Although most studies have concentrated on the composition of fecal bacteria, early indications suggest that the gut microbiome’s variability varies in different regions of the gastrointestinal tract [93,94]. Importantly, intestinal resections, to some extent, reduce the diversity of microbiota present in the remaining gut and colon [95,96]. This change triggers an increase in the prevalence of certain Gram-positive bacterial communities, such as facultative anaerobes like Lactobacillus [97].

## 4. Potential Impact of Parenteral Nutrition on Gut Barrier and Microbiota

Numerous research studies have explored the link between nutrition and the intestinal barrier, particularly in the context of inflammatory bowel disease. Exclusive enteral nutrition has been shown to effectively induce remission of CD in children and is now recommended as a primary treatment option in active (mild to moderate) pediatric CD [98]. The precise reason for the success of enteral nutrition is not yet entirely clear, but it could be attributed to preventing exposure to dietary antigens and specific food components by bringing about changes in the gut barrier and microbiota. On the other side, when enteral feeding is not possible, parenteral nutrition can prevent malnutrition and support intestinal failure.

However, there is a lack of definite information on the potential negative impact of parenteral nutrition on the intestinal barrier. Some preclinical and clinical studies have suggested that PN may have a negative effect in this regard (Figure 3).

### 4.1. Intestinal Atrophic Damage and Role of Intestinotrophic Agents

Preclinical studies suggest that parenteral nutrition can cause atrophic damage to the intestinal mucosa.

In a study in porcine models, a reduction in muscle mucosal thickness and marked atrophy of the small intestine was observed after total parenteral nutrition, compared with enteral nutrition. Specifically, significant atrophy of intestinal villi and a reduction in intestinal ‘mass’ related to a loss in the muscle layer was demonstrated [99]. Mucosal atrophy can occur in the context of PN and is characterized by a decrease in villus height, crypt depth, surface area, and number of epithelial cells, which can occur as early as 24 h after the onset of PN [100]. Similarly, a morphometric analysis of the distal small intestine of newborn pigs revealed significantly reduced villus blunting and villus/crypt ratio in the TPN-exposed group for a longer time duration of 14 days, compared with the EN group [101]. This relatively rapid onset of mucosal atrophy may be due to the high metabolic rate and rapid turnover of intestinal epithelial cells without their optimal replacement [102]. Atrophy in terms of markedly reduced circumference throughout the intestinal tract was also observed in rats subjected to TPN for 8 days. A reduction in the width of the jejunal and ileal villi and in the length of the jejunal villi was also observed; in contrast, the thickness of the submucosa in the jejunum and ileum and the number of Paneth cells increased [103]. Further studies in porcine and murine models have confirmed these aspects of small intestinal mucosal atrophy after total parenteral nutrition compared with animals fed enteral nutrition [99,103,104].

As mentioned earlier for inflammatory bowel disease, enteral or oral nutrition has an important influence on intestinal mucosal health. Another possible theory postulated is that it is indeed the presence of luminal chyme, as well as intestinal motility, that positively influences intestinal mucosal trophism. Indeed, it has been shown on human intestinal cell lines that the absence of nutrients in the lumen and mucosal contact with luminal chyme play a role in intestinal villus hypoplasia [105]. On the other hand, PN may result in greater synergistic damage, with additional starvation effects on intestinal mucosa [106].

In an effort to reduce the intestinal atrophy caused by parenteral nutrition (PN), researchers have studied the efficacy of intestinotrophic agents in weaning patients off of PN. PN has been linked to reduced production of certain hormones, making the use of such agents all the more crucial [101,107]. This hormone deficiency is thought to result from enteral starvation mechanisms. For instance, glucagon-like peptide-2 (GLP-2) is released when bile acids activate the TGR5 receptor in the ileum, but this process is disrupted when intestinal function is compromised [108]. The United States has approved the use of two agents—recombinant growth hormone (GH) and teduglutide, a GLP-2 analog—for weaning patients with SBS-IF off of PN. These agents have also been found to be beneficial for patients with inflammatory bowel disease [109]. Although some randomized clinical trials have shown discrete weaning rates from PN, recombinant GH has not been widely accepted in clinical practice due to concerns about its efficacy and adverse effects. The conflicting results regarding nutrient and fluid absorption in humans have also contributed to these doubts [110,111,112].

The real revolution has occurred since the introduction of GLP-2 analogs. GLP-2, an intestinal growth-promoting factor, is a 33-amino-acid peptide derived from enteroendocrine L cells of the intestinal epithelium [113] that directly or indirectly stimulates intestinal nutrient absorption and growth [114,115]. The role of GLP-2 as a growth modulator of epithelial cells has been demonstrated in mouse models of short bowel syndrome. It appears to induce the proliferation of stem cells and leads to the lengthening of crypts and enhanced epithelial responses [116]. It is worth noting that the way teduglutide works is quite complicated and involves both direct and indirect integration effects with a GLP-2 receptor. The most significant factors are as follows: it promotes the proliferation of crypt cells, enhances intestinal weight and villus growth, improves intestinal barrier function, inhibits gastrointestinal tract motility and gastric acid secretion, and increases intestinal blood flow [117]. The dipeptidase-resistant GLP-2 analog (teduglutide) studied in patients with short bowel syndrome resulted in histologic increases in villus height and mitotic index [118,119]. From early clinical trials to real-world studies, teduglutide has demonstrated relevant clinical utility for patients with SBS-IF. It has significantly reduced HPN volume and/or infusion days, allowing enteral autonomy in some patients. Moreover, it has shown some reassuring data in terms of efficacy and tolerability, even in Crohn’s disease [120].

### 4.2. Chemical and Immunological Barrier Hazard

Mucosal atrophy is often associated with decreased GALT cellularity and vascular perfusion, which are, in turn, related, as naïve lymphocytes normally circulate systemically and interact with ligands when they enter the splanchnic vascular system [121].

In mice, the total number of Peyer’s patch lymphocytes decreases significantly within 24–48 h after the onset of PN, reaching a nadir of 50–75% reduction within 3 days, compared with control animals [122].

Interestingly, no changes occur in the ratios of total T lymphocytes to B lymphocytes and in the relative population of the various subgroups. Specifically, the ratio of CD4+ to CD8+, as well as the relative percentage of memory cells, activated cells, and naïve cells within the T- and B-lymphocyte populations, remains stable in Peyer’s patches [123]. Studies also show that the primary outcome of PN on Peyer’s patches is reduced cell recruitment, fewer mucosa-specific activated lymphocytes, and reduced lymphoid signaling. Furthermore, enteral re-feeding after PN reciprocally restores normal Peyer’s patch cellularity within 48 h, demonstrating that these changes are sensitive and reversible [124]. Again, PN significantly reduces the total number of lymphocytes in the lamina propria and induces important changes in CD4+ lymphocyte population subsets that affect the production of immunoglobulin A (IgA) and antimicrobial products. For example, the reduction in Th2 cells that is observed after PN and in related cytokines, such as IL-4, IL-5, IL-6, IL-9, IL-10, IL-13, and IL-25, generates a reduction in IgA production in plasma cells and IgA transport in epithelial cells [125].

In this context, reduced levels of IL-4 and IL-13 are associated with reduced Paneth cell gene expression of lysozyme, sPLA 2, cryptidin-4, and RegIIIγ [126,127], which are associated with decreased luminal levels of Paneth cell products [128]. After PN, no changes in the total number of calyciform cells were observed; however, gene expression and luminal levels of MUC2 and trefoil factor 3 (TFF3, epithelial repair promoter) decreased in pigs and newborn mice [48,129]. In addition, gene expression levels of RELMβ protein, which has a chemotactic role in CD4+ T lymphocytes and in stimulating mucus and antimicrobial production [130], were found to be reduced following PN [129]. Interestingly, experimental administration of the Th2-stimulating cytokine IL-25 during PN maintained luminal antimicrobial levels and increased the number of calyciform cells and tissue and luminal MUC2 levels compared with controls or PN alone [127,131]. In practical terms, these observed functional changes affect susceptibility to pathogens. As observed, the small intestine of PN-fed animals released lower levels of antimicrobial compounds compared with enteral controls, which resulted in a reduced ability to kill colony-forming units (CFUs) of Pseudomonas aeruginosa in vitro [132]. Furthermore, in an ex vivo intestinal segment culture, PN administration resulted in decreased mucosal sPLA2 release and a significant increase in enteroinvasive E. coli colonies in less than 60 min, compared with tissue from control animals [133].

### 4.3. Gut Dysbiosis

Research indicates that parenteral nutrition can result in considerable changes to the gut microbial population (Table 1).

Early evidence was derived from animal studies, which prompted researchers to conduct several investigations to examine the impact of PN and lack of enteral nutrition on gut microbiota. A study conducted by Hodin et al. on rats showed that after 14 days of PN, there was a significant shift in the microbial representation. The abundance of Firmicutes decreased, while the abundance of Bacteroidetes remained unchanged, causing the microbial representation to shift in favor of Bacteroidetes [134]. Two studies conducted on mouse models showed that after 5 days of PN, the dominant phyla in the small intestine of mice shifted to Proteobacteria and Bacteroidetes from the previously dominant phylum Firmicutes [135,136]. At the genus level, PN mice had higher levels of bacteria in Salmonella, Escherichia, Proteus, and Bacteroides, which are often associated with clinical infections [135].

Experimental studies were conducted on controlled animal models to investigate the impact of PN on neonatal intestinal colonization. The results showed that piglets fed via PN had lower bacterial diversity, higher bacterial concentrations (CFU/g), and lower colonization of all segments of the intestinal tract as compared to the enterally fed controls. Moreover, PN-fed piglets showed a higher colonization rate by toxin-expressing strains of C. difficile [137]. Another study conducted by Deplancke et al. found that the bacterial community structure in the ileum of piglets fed enterally and parenterally was equally complex. However, after seven days of parenteral nutrition, the microbiota of piglets showed an increase in bacteria that could utilize sulfated monosaccharides and mucus-associated bacteria [138]. Furthermore, Lavallee et al. published a study that found that the composition of the microbiome in infants is not only influenced by parenteral nutrition (PN) itself but also by the type of lipid constituent. The group that received PN with prevalent ω-6 polyunsaturated fatty acids (PUFAs) had a specific and significant increase in Parabacteroides, while the group that received PN with ω-3 PUFAs showed an increase in Enterobacteriaceae [139].

Preclinical studies have provided some early data that are supported by initial evidence in both adults and children suffering from short bowel syndrome. Studies conducted on SBS patients who rely on parenteral nutrition have produced consistent findings that are similar between both adults and children. The most consistent result across these studies is a decline in bacterial diversity [140,141,142,143,144], which appears to be linked to the length of the remaining small intestine [141] and the duration of PN dependence [143]. Additionally, a few studies have noted a decrease in bacterial richness, which refers to the number of species present in a microbial community. Through the analysis of the microbial composition of individuals who have undergone parenteral nutrition treatment, numerous studies have found a notable rise in Gram-negative proteobacteria, with a particular emphasis on gammaproteobacteria and their Enterobacteriaceae family [84,140,141,142,143,144]. This increase in proteobacteria is a result of the absence of fermentable food substrates, such as fiber and resistant starch, in the intestinal lumen [145].

In addition, a decrease in Bacteroidetes has been observed [85,140,141,142], and patients with intestinal failure exhibited low levels of Firmicutes of the order Clostridiales [140,142,144]. The microbiota of these patients, while maintaining good levels of Firmicutes, was depleted in anaerobic microorganisms, particularly those of the family Clostridiaceae and some important butyrate-producing families, such as Lachnospiraceae, Ruminococcaceae, and others [146]. Some studies have found that patients with intestinal failure or low levels of Firmicutes, which are major fiber fermenters, also have low levels of butyrate-producing bacteria [140,147,148]. Butyrate and SCFA deficiencies affect the established anti-inflammatory and energetic properties of the colonic intestinal epithelium of these metabolites. Moreover, it has been observed that individuals who are not healthy have a higher amount of bacilli, especially lactobacilli, as compared to healthy individuals. The consequences of this increase in lactobacilli are not yet well understood, but some initial evidence suggests that certain strains of lactobacilli might be linked to either shorter or longer duration of PN. Additionally, some strains of lactobacilli might also be associated with conditions like diarrhea and D-lactic acidosis [149]. A 2019 study by Budniska et al. revealed that, besides shared characteristics and species among all categories exposed to PN, each SBS type had a unique composition. The study categorized SBS into SBS I (jejunostomy), SBS II (jejunocolonic), and non-PN SBS (jejunocolic without TPN dependence), demonstrating different microflora compositions for each classification. SBS I patients exhibited a high load of aerobic bacteria with concurrent depletion of anaerobes. The study also observed some normalization of the fecal microbiome in non-TPN SBS patients. Notably, all SBS patients exhibited high medium-chain fatty acids and saturated aldehydes and reduced short-chain fatty acids. Moreover, they were enriched by chenodeoxycholic and deoxycholic acid while being depleted of lithocholic acid [150].

**Table 1 nutrients-16-02288-t001:** Major studies examining parenteral nutrition’s impact on gut microbiota.

TYPE OF STUDY	STUDY DESIGN	METHOD	RESULTS	REFERENCES
Animals	Rats receiving TPN for 3, 7, or 14 days compared to control rats	Ileal lumen contents and ileal biopsies	↓↓ Firmicutes in TPN group No significant differences in *Bacteroidetes*	Hodin et al. (2012) [134]
Animals	WT or MyD88−/− mice receiving PN solution for 5 days Controls with free access to chow	Segments of small intestine and colon preservation of mucosa-associated bacteria	Domination of *Proteobacteria* and *Bacteroidetes* in the TPN group ↑↑ *Salmonella*, *Escherichia*, *Proteus* and *Bacteroides*	Miyasaka et al. (2013) [135]
Animals	Male mice randomized to chow or PN for 5 days	Segments of ileum 16S rRNA sequencing	PN group: ↑↑ *Bacteroidetes* and *Lactobacillus* ↓↓ *Firmicutes*	Heneghan et al. (2013) [136]
Animals	Neonatal piglets with PN for 7 days Piglets with EN Orally fed piglets	Mucosal samples of the ileum	TPN ileum was enriched in mucolytic bacteria ↑ *C. perfringens* in the TPN ileum than in the TEN ileum	Deplancke et al. (2002) [138]
Animals	Neonatal piglets with TPN for 14 days Healthy controls	Microbial composition in ileal mucosa 16S rRNA sequencing	Mixed Lipid piglets more similar to Soybean Oil ↑ *Parabacteroides* with Soybean Oil ↑ *Enterobacteriaceae* with Mixed Lipid	Lavallee et al. (2017) [139]
Humans	Children with IF - on PN after 57 months of PN - weaned off PN Healthy controls	Fecal samples Culture-independent phylogenetic DNA-based microarray analysis	↓ Diversity and richness ↑ *Lactobacilli*, *Proteobacteria and Actinobacteria*, *Clostridium* clusters IX, XIII, and XV, *Fusobacteria*, *Spirochetes* ↓ Clostridium clusters III, IV, and XIVa	Korpela et al. (2017) [144]
Humans	SBS children weaned from parenteral SBS children on PN therapy + oral and/or enteral intake Healthy controls	Fecal samples 16S rRNA sequencing	↓ Bacterial diversity in children with SBS receiving PN vs. children weaned off PN Patients with suspected SIBO: ↑ Enterobacteriaceae, patient receiving PN without suspected SIBO: ↑ Lactobacillaceae	Lilja et al. (2015) [143]
Humans	Children with SBS dependent on PN Healthy controls	Fecal samples 16S rRNA sequencing Measurement of SCFAs	↓ Richness in all SBS groups ↓ Acetate in SBS groups Equal propionate and butyrate and total SCFAs	Wang et al. (2017) [140]
Humans	Adults with SBS on PN 24 months after the final digestive circuit modification Healthy controls	Fecal samples, culture-dependent method	↓ Bacterial counts ↓ Bacteroidetes, Firmicutes, Bifidobacterium, and *Methanobrevibacter smithii*	Boccia et al. (2017) [85]
Humans	SBS children dependent on PN Children weaned from PN Healthy control subjects	Fecal samples 16S rRNA sequencing	↓ Firmicutes order Clostridiales ↓ Firmicutes and ↑ Enterobacteriaceae in SBS group (and poor growth)	Piper et al. (2017) [148]

PN: parenteral nutrition; SBS: short bowel syndrome; EN: enteral nutrition; TPN: total parenteral nutrition; TEN: total enteral nutrition; SCFA: short-chain fatty acids; WT: wild-type. up arrow: increase, down arrow: decrease.

### 4.4. Epithelial Barrier and Intestinal Permeability

The gut needs a strong epithelial barrier function to protect against harmful substances, toxins, antigens, and bacteria while also allowing for efficient nutrient absorption. Parenteral nutrition (PN) can cause the loss of this barrier function, leading to increased intestinal permeability. This is due to several factors, including gut atrophy, that cause damage by shrinking small intestinal villi, increasing cell death, and decreasing cell growth [151,152,153]. Additionally, PN administration can trigger the production of proinflammatory cytokines [154,155], leading to further damage and increased intestinal permeability [156].

Researchers used mouse models to examine how total parenteral nutrition (TPN) affects intestinal barrier function and the underlying mechanisms [151,155,157,158]. These studies revealed that TPN reliance triggers a proinflammatory response in the intestine, leading to an increase in intraepithelial lymphocyte (IEL)-derived TNF-α expression. This increase plays a crucial role in the loss of intestinal barrier function and tight junction proteins in TPN mice [154,159]. Additionally, these studies found that the augmented expression of IFN-γ by IELs, coupled with a reduced production of IL-10, also contributes to the loss of the epithelial barrier associated with TPN [158,160,161]. According to this research, prolonged parenteral nutrition (PN) can diminish the expression of vital intestinal tight junction proteins, such as zonula occludens 1, E-cadherin, and claudins. These proteins have a significant role in preserving the intestinal barrier and regulating ion transport. When their regulation is disrupted, the intestinal barrier’s function is compromised, leading to the translocation of microbial products [158]. Such translocation is a crucial factor in the emergence of septic complications associated with PN reliance [153,162,163].

A damaged barrier also appears to play a key role in the development of a liver disorder known as Parenteral Nutrition-Associated Liver Disease (PNALD). The ‘dose-dependent’ toxicity of nutrients such as glucose, ω-6 fatty acids, phytosterols, and some trace elements represents one of the major mechanisms of liver damage inherent in parenteral nutrition itself. However, EN deficiency impairs the secretion of gastrointestinal hormones (gastrin, motilin, secretin, and glucagon, among others), thus leading to major abnormalities in intestinal motility, gallbladder contractility, enterohepatic circulation, and bile acid secretion/absorption, potentially increasing the risk of cholestasis and subsequent PNALD [164,165]. In this context, changes in the intestinal microbiome and reduced epithelial and chemical barrier function may result in altered intestinal permeability and the release of proinflammatory cytokines by immune cells, leading to the translocation of harmful compounds such as LPS into the liver [166]. Increased bacterial translocation characterized by increased bacterial counts in the cecum and mesenteric lymph nodes was detected in rats after total parenteral nutrition for 2 weeks [167]. Rats treated with sodium dextran sulfate, followed by TPN for 7 days, showed increased intestinal permeability and portal LPS levels, Kupffer cell (KC) activation, hepatocyte damage, and cholestasis. Interestingly, although EN can normalize cholestasis, improvements in liver function generally do not begin until full EN is tolerated and PN is discontinued [102].

## 5. Conclusions, Challenges, and Future Perspectives

Malnutrition is a significant and often undetected problem for people with inflammatory bowel disease (IBD), which can negatively impact medical treatment, surgical outcomes, and overall well-being [168]. Enteral nutrition is the preferred approach to improving nutrition, although, in specific scenarios, such as in patients with short bowel syndrome, high-output gastrointestinal fistula, or complete small bowel obstruction, parenteral nutrition (PN) is the recommended course of action [28]. In recent years, there has been a growing interest in exploring the potential adverse effects of parenteral nutrition on the gut barrier and the composition and function of the gut microbiota. Parenteral nutrition is known to lead to significant changes in the gut barrier ecosystem, including immune–microbiological dysregulation, epithelial damage, and alterations in gut barrier permeability [151,152,153]. However, the impact of parenteral nutrition on the intestinal barrier in IBD, which already has a compromised barrier, is not yet fully understood.

Indeed, the evidence collected to date has involved both humans and animal models, many of which are, however, healthy or affected by SBS. Nevertheless, there appear to be interrelationships between short- and long-term PN and various negative impacts on the gut barrier, with aspects generally overlapping with the barrier alterations we observe in IBD. In recent preclinical studies, it has been found that the administration of total parenteral nutrition (TPN) provokes a proinflammatory reaction in the intestine, resulting in the heightened production of TNF-a by intraepithelial lymphocytes (IELs). This, in turn, hampers the function of the tight junctions as well as disrupts the gut barrier [154,159]. The elevated intestinal permeability associated with TPN also appears to be linked to other factors, such as intestinal atrophy and decreased cell growth [151,152,153]. In IBD, heightened intestinal permeability is primarily linked to irregular tight junctions and extensive mucosal damage during active disease phases, which further impairs barrier function [4]. Within the web of proinflammatory cytokines central to IBD pathophysiology, TNF-alpha, among others, has been implicated in tight junction disruption and epithelial cell apoptosis. Moreover, it is now understood that gut dysbiosis is closely linked to heightened intestinal permeability. This occurs through processes involving inflammation, changes in metabolite production, and immune system modulation [70]. The types of microbial imbalance seen in individuals with inflammatory bowel disease (IBD) and TPN-exposed SBS patients involve a decrease in microbial diversity and an increase in bacteria from the phylum Proteobacteria [63,84,140,141,142,143,144]. Many of these bacteria play a proinflammatory role. Nonetheless, when examining the specific microbial signatures in the two conditions, they differ in terms of reduced richness and alteration in the relative abundance of certain bacterial taxa.

Unfortunately, it is still unclear how parenteral nutrition can influence the gut barrier and the altered microbiota in IBD patients. The analyzed studies have limitations related to small sample sizes and the heterogeneous characteristics of the population in clinical trials. Additionally, there is often a lack of information regarding the amount and type of oral nutrition/enteral nutrition used. Hence, it is imperative to thoroughly investigate the interactions of the gut microbiota and gut barrier integrity during the use of parenteral nutrition in order to comprehend the mechanisms involved and develop preventive approaches to mitigate potential damage to the gut system. Continued research on the role of prebiotics, probiotics, symbionts, and other therapeutic strategies is essential to identify new ways to enhance gut health and overall well-being. Incorporating human models analyzed with a multiomics approach is necessary to pave the way for large, well-designed, randomized controlled trials and significantly expand knowledge in this crucial area. In conclusion, given the significance of parenteral nutrition in IBD, there is an urgent need to clarify its actual impact on the barrier in these patients. Additionally, it is important to understand how these effects may influence clinical decisions and outcomes, potentially leading to the development of new therapeutic strategies.

## Figures and Tables

**Figure 1 nutrients-16-02288-f001:**
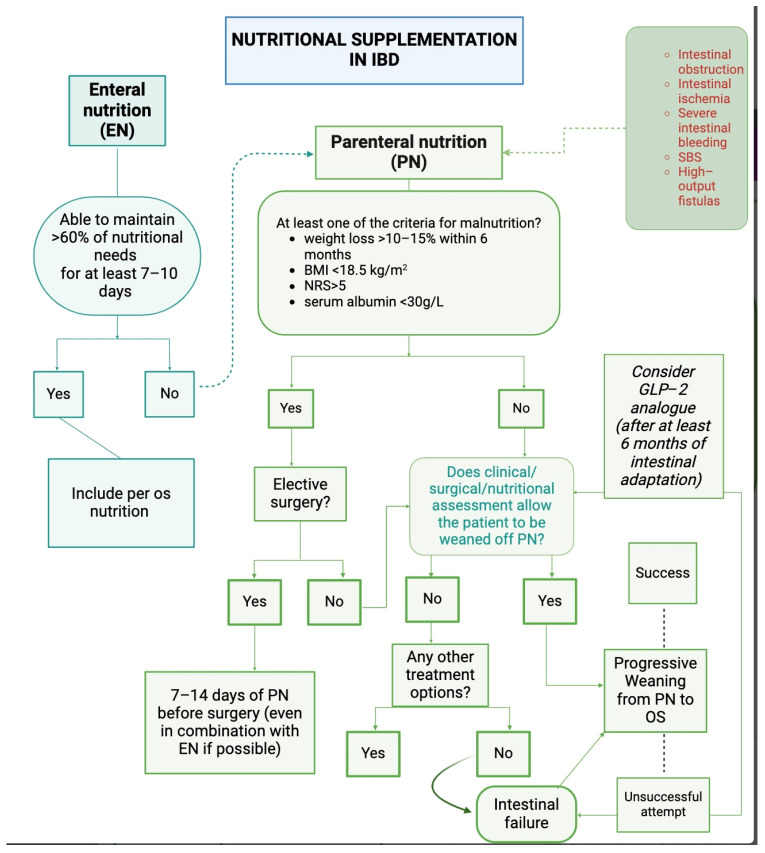
Example of algorithm for management of nutritional support in inflammatory bowel disease.

**Figure 2 nutrients-16-02288-f002:**
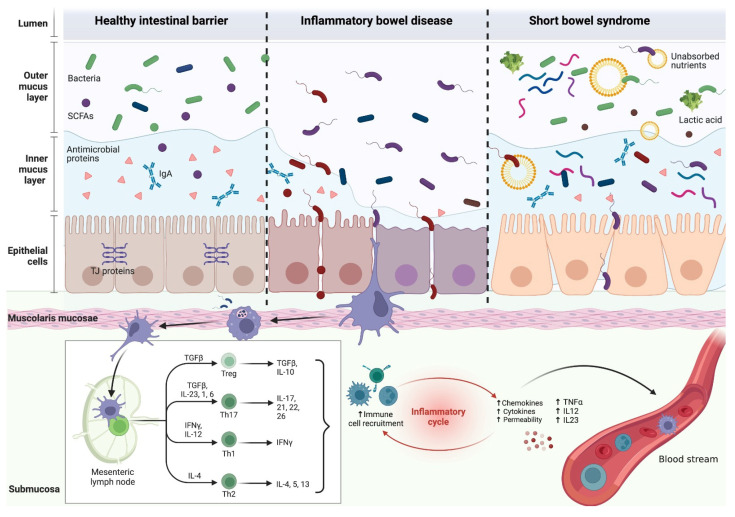
Gut barrier disruption in inflammatory bowel disease and short bowel syndrome.

**Figure 3 nutrients-16-02288-f003:**
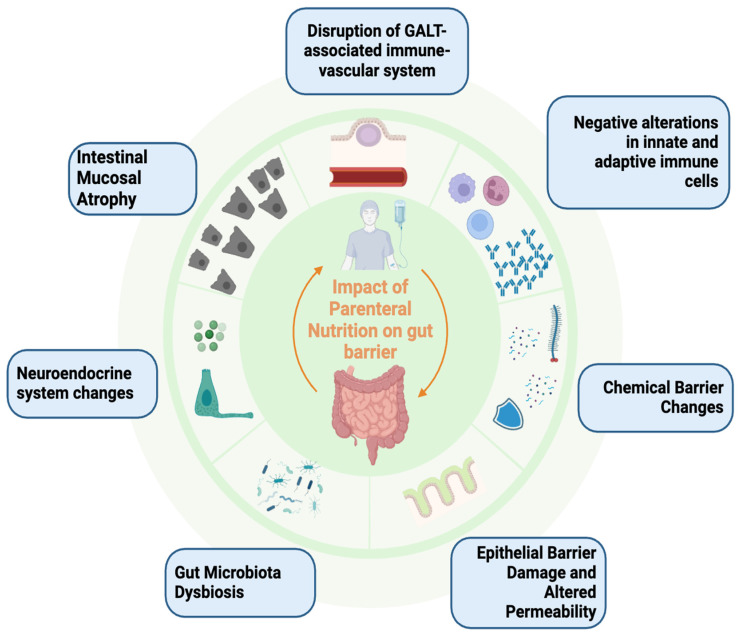
Overview of the impact of parenteral nutrition on gut barrier.

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
