# Peer review of "Parenteral Nutrition, Inflammatory Bowel Disease, and Gut Barrier: An Intricate Plot"

_nutrients, 2024, doi:10.3390/nu16142288_

Round 1
Reviewer 1 Report
Comments and Suggestions for Authors
This review summarizes the parenteral nutrition in IBD. It seem written by inexperienced authors who don’t know how to handle sentences and paragraphs, common knowledge and new findings, general points and specific ones. It is just too substantial to have this manuscript revised such it can meet the criteria for publication.
1. There are too many paragraphs with less than 10 lines of words, which looks very unprofessional. The authors should seriously think about combining these short paragraphs into longer ones.
2. For all the figures, figure title should be placed below the figure.
3. Abbreviations should be defined at their first appearance in the manuscript, for example, ESPEN in line 127.
4. As this is supposed to be a review paper for specialized experts, general knowledge should be deleted, such as the introduction of intestinal failure, human gut microbiota and barrier, and so on.
Comments on the Quality of English LanguageThe authors should work extensively in English
Author Response
Dear editorial assistant and reviewers,
We want to express our gratitude for the opportunity to work on this manuscript. We have addressed the feedback provided by the reviewers and made the necessary changes to the content.
The modifications made to the manuscript include:
- Refining the chapter 'Parenteral Nutrition in Inflammatory Bowel Disease' by removing two subsections, streamlining the content, and aligning it with an illustrative management algorithm (which has also been modified).
- Streamlining the general evidence on the barrier and gut microbiota to underscore their significance in understanding the main text
- Revising the conclusions to highlight the most critical aspects and provide insights into future perspectives
- Making subtle adjustments to the layout, abbreviations, and language usage where deemed appropriate.
Reviewer #1
1. There are too many paragraphs with less than 10 lines of words, which looks very unprofessional. The authors should seriously think about combining these short paragraphs into longer ones.
Response: We thank the reviewer for the comment. We modified it as suggested.
2. For all the figures, figure title should be placed below the figure.
Response: We thank the reviewer for the comment. We modified it as suggested.
3. Abbreviations should be defined at their first appearance in the manuscript, for example, ESPEN in line 127.
Response: We thank the reviewer for the clarification. We performed a check on the abbreviations.
4. As this is supposed to be a review paper for specialized experts, general knowledge should be deleted, such as the introduction of intestinal failure, human gut microbiota and barrier, and so on.
Response: We thank the reviewer for the suggestion. Having considered the relevance of the introduction about intestinal failure, human gut microbiota, and barrier in comprehending the text, we agree that a more streamlined approach to this section, with significant modifications, would enhance the readability of the manuscript. We have made adjustments to our approach, ensuring that even individuals without expertise in the subject matter can follow along with the main content as clearly as possible.
Reviewer 2 Report
Comments and Suggestions for Authors
The authors have submitted a well written, extremely comprehensive review on the use of parenteral nutrition in inflammatory bowel diseases and short bowel syndrome, and the impact of parenteral nutrition on the gut. It was an easy read.
However, the title is somewhat misleading. The impression is that the authors will explore the impact of parenteral nutrition on the gut of IBD patients, however it isn’t so. This is, of course, due to lack of literature on the subject. The title of the paper should be changed to reflect what it really is about, the impact of PN and IBD on the gut and the changes PN could have on the already dysfunctional gut of IBD patients. Furthermore, in the discussion the authors should reflect deeper on the similar or dissimilar impacts of PN and IBD on the gut, the implications that combining the two could have on a dysfunctional gut, and pinpoint specific aspects that need further studies. The discussion is a bit vague as it is and needs to be re-written.
The authors state that figure 1 is the algorithm for correct nutritional support for IBD; according to whom? References are needed here.
The authors state that PN should be started within 24h after a surgery. However the authors do not cite articles with relation to PN and surgery. The 2009 ESPEN guidelines recommend between 7-10 days postop or 5-7 in patients already malnourished preoperatively.
Although a great summary, the whole introduction of chapter 3 is much too long and should be drastically shortened as it is not really the subject of the paper.
The authors state, on page 11 line 441, that figure 2 shows the systemic consequences of lactic acid buildup in SBS. That is not so. Perhaps the authors should add this to the figure, in addition to the systemic effects in IBD.
The paragraph on GLP-2 analog does not seem to be in the correct spot in the paper. The authors should check this for a smoother read.
All abbreviations should be introduced in full text, in the text and table 1 (what are ML, SF, SO?).
Comments on the Quality of English LanguageOverall the english is good.
Please recheck the document for minor mistakes where words are in Italian, capital letters that are not at the start of a phrase, punctuation.
What is the digiunal villus mentioned twice in the document?
Author Response
Dear editorial assistant and reviewers,
We want to express our gratitude for the opportunity to work on this manuscript. We have addressed the feedback provided by the reviewers and made the necessary changes to the content.
The modifications made to the manuscript include:
- Refining the chapter 'Parenteral Nutrition in Inflammatory Bowel Disease' by removing two subsections, streamlining the content, and aligning it with an illustrative management algorithm (which has also been modified).
- Streamlining the general evidence on the barrier and gut microbiota to underscore their significance in understanding the main text
- Revising the conclusions to highlight the most critical aspects and provide insights into future perspectives
- Making subtle adjustments to the layout, abbreviations, and language usage where deemed appropriate.
Reviewer #2
1. However, the title is somewhat misleading. The impression is that the authors will explore the impact of parenteral nutrition on the gut of IBD patients, however it isn’t so. This is, of course, due to lack of literature on the subject. The title of the paper should be changed to reflect what it really is about, the impact of PN and IBD on the gut and the changes PN could have on the already dysfunctional gut of IBD patients. Furthermore, in the discussion the authors should reflect deeper on the similar or dissimilar impacts of PN and IBD on the gut, the implications that combining the two could have on a dysfunctional gut, and pinpoint specific aspects that need further studies. The discussion is a bit vague as it is and needs to be re-written.
Response: We appreciate the feedback from the reviewer. As per your suggestion, we have updated the title "Parenteral Nutrition, Inflammatory Bowel Disease, and Gut Barrier: An Intricate Plot" to better reflect the content and pique the reader's interest. Additionally, we have revised the discussion to emphasize the key aspects you highlighted. Thank you for your valuable input.
2. The authors state that figure 1 is the algorithm for correct nutritional support for IBD; according to whom? References are needed here.
Response: We appreciate the reviewer for providing clarification. The proposed algorithm serves as an example of nutritional support management for IBD patients. We have made revisions, particularly to the section on parenteral nutrition in IBD, to ensure it flows more smoothly and aligns with the example in the image. Consequently, you will also find the references included in the text. Thank you for allowing us to address this point.
3. The authors state that PN should be started within 24h after a surgery. However the authors do not cite articles with relation to PN and surgery. The 2009 ESPEN guidelines recommend between 7-10 days postop or 5-7 in patients already malnourished preoperatively.
Response: We thank the reviewer for the observation. However, the ESPEN 2021 guidelines for clinical nutrition in surgery recommend the initiation of enteral nutrition within 24 hours postoperatively. We will provide to insert the citation and thank the reviewer again (RECOMMENDATION 24). According to your comment, we provided to specify that PN should be administered as soon as possible when enteral feeding is not possible, in accordance with the ESPEN guidelines 2021.
4. Although a great summary, the whole introduction of chapter 3 is much too long and should be drastically shortened as it is not really the subject of the paper.
Response: We thank the reviewer for the suggestion. Having considered the relevance of the general introduction about human gut microbiota, and barrier in comprehending the text, we agree that a more streamlined approach to this section, with significant modifications, would enhance the readability of the manuscript. We have made adjustments to our approach, ensuring that even individuals without expertise in the subject matter can follow along with the main content as clearly as possible.
Round 2
Reviewer 1 Report
Comments and Suggestions for Authors
The authors have replaced many references with the same reference and the reviewer feels very confused about this.
Comments on the Quality of English LanguageThe English can be further improved
Author Response
Dear editorial assistant and reviewers,
below are the clarifications and responses to the comments made by the reviewers.
Reviewer #1
The authors have replaced many references with the same reference and the reviewer feels very confused about this.
Response: We have made specific adjustments to the references in the text to ensure complete clarity for the reviewer.
Reviewer 2 Report
Comments and Suggestions for Authors
Thank you to the authors for making the requested changes, notably in the title and Chapter 3. The paper is much more readable, and the discussion does now underline much needed missing data in IBD and PN.
However, there are some repeating segments in Chapter 3 that need removal (ie, the start of paragraph line 287 page 9, and line 369 page 10)
A question:
Figure 1. Why is the recommendation for IBD patients under PN to undergo TPN prior to surgery? If possible, a combination of EN and PN should be recommended in these patients. If it is a mistake, please correct in Figure 1 and in the text. If not, please explain the reasoning behind TPN.
A few suggestions:
- Remove sub-chapter title 2.1, as there are no other subchapters in this segment.
- Add sub-chapter title 3.1 "Composition and function of a healthy gut"; to differentiate from sub-chapter "Specific barrier changes in..."
- Line 148, page 5: SBS is written without first introducing the abbreviation
- Line 367, page 10: I think the authors meant "dynamic" microorganism communities
- The discussion needs references in parts that state known "facts".
Author Response
REVISIONE 2
Dear editorial assistant and reviewers,
below are the clarifications and responses to the comments made by the reviewers.
Reviewer #1
The authors have replaced many references with the same reference and the reviewer feels very confused about this.
Response: We have made specific adjustments to the references in the text to ensure complete clarity for the reviewer.
Reviewer #2
Thank you to the authors for making the requested changes, notably in the title and Chapter 3. The paper is much more readable, and the discussion does now underline much needed missing data in IBD and PN.
-However, there are some repeating segments in Chapter 3 that need removal (ie, the start of paragraph line 287 page 9, and line 369 page 10)
Response: We thank the reviewer for the comment. We modified the text as suggested.
-A question:
Figure 1. Why is the recommendation for IBD patients under PN to undergo TPN prior to surgery? If possible, a combination of EN and PN should be recommended in these patients. If it is a mistake, please correct in Figure 1 and in the text. If not, please explain the reasoning behind TPN.
Response: We thank the reviewer for the intelligent remark. Indeed, the way the proposed algorithm example was designed and from the recommendations provided by the guidelines, it would be more correct to specify that PN appears to be essential in the nutritional support of malnourished patients before surgery, but under conditions that may allow it, a combination with enteral support is recommended. We have modified the image and text accordingly.
-A few suggestions:
- Remove sub-chapter title 2.1, as there are no other subchapters in this segment.
- Add sub-chapter title 3.1 "Composition and function of a healthy gut"; to differentiate from sub-chapter "Specific barrier changes in..."
- Line 148, page 5: SBS is written without first introducing the abbreviation
- Line 367, page 10: I think the authors meant "dynamic" microorganism communities
- The discussion needs references in parts that state known "facts".
Response: We thank the reviewer for the comment. We modified the text as suggested.